# Antiobesity and Antioxidative Effect of Fermented Brown Rice Using In Vitro with In Vivo *Caenorhabditis elegans* Model

**DOI:** 10.3390/life13020374

**Published:** 2023-01-29

**Authors:** Kaliyan Barathikannan, Akanksha Tyagi, Lingyue Shan, Nam-Hyeon Kim, Deuk-Sik Lee, Jong-Soon Park, Ramachandran Chelliah, Deog-Hwan Oh

**Affiliations:** 1Department of Food Science and Biotechnology, College of Agriculture and Life Sciences, Kangwon National University, Chuncheon 200701, Republic of Korea; 2Agricultural and Life Science Research Institute, Kangwon National University, Chuncheon 24341, Republic of Korea; 3Life Science Institute, Well-being LS Co., Ltd., Gangneung 25451, Republic of Korea; 4Kangwon Institute of Inclusive Technology KIIT, Kangwon National University, Chuncheon 24341, Republic of Korea; 5Saveetha School of Engineering, SIMATS, Chennai, Tamil Nadu 600124, India

**Keywords:** brown rice, *Pediococcus acidilactici* MNL5, lipase inhibitory, ferulic acid, lipid reduction

## Abstract

Naturally occurring phytochemicals from plants or grains are crucial in reducing various metabolic disorders. Bioactive phytonutrients are abundant in the Asian dietary staple, brown rice. This research evaluated the impact of lactic acid bacteria (LABs) bioconversion and fermentation on antioxidant and antiobesity activities and ferulic acid content in brown rice. The combination of bioconversion with *Pediococcus acidilactici* MNL5 among all LABs used showed a synergistic impact with 24 h of solid-state brown rice fermentation. The 24-h MNL5 fermented brown rice (FBR) demonstrated the most potent pancreatic lipase inhibitory activity (85.5 ± 1.25%) compared to raw brown rice (RBR) (54.4 ± 0.86%). The antioxidant potential of MNL5-FBR was also found to be highest in the DPPH assay (124.40 ± 2.40 mg Trolox Equiv./100 g, DW), ABTS assay (130.52 ± 2.32 mg Trolox Equiv./100 g, DW), and FRAP assay (116.16 ± 2.42 mg Trolox Equiv./100 g, DW). Based on higher antioxidant and antiobesity activities, samples were quantified for ferulic acid content using the HPLC-MS/MS approach. Furthermore, *C. elegans* supplementation with FBR showed enhanced life span and lipid reduction in fluorescence microscope analysis compared to the control. Our results indicate that the expression study using the *C. elegans* model (N2 and Daf-2 models) fat gene was conducted, showing a lowering of obesity ability in FBR-fed worms. Our study indicates that FBR has improved antioxidant and antiobesity actions, especially in MNL5-FBR, and can be employed to develop functional foods that combat obesity.

## 1. Introduction

Obesity affects almost one billion people worldwide, including 650 million adults, 340 million adolescents, and 39 million toddlers, and the figure is growing every day. According to the WHO, by 2025, approximately 167 million individuals, including adults and youngsters, will become less healthy due to becoming overweight or obese [1]. Diet is the most significant health factor in reducing obesity and is directly linked to microbiome regulation. Numerous studies have demonstrated that genetic vulnerability to obesity can be linked to an obesogenic environment (e.g., a drastic change in food that affects the gut flora, lack of exercise, and a poor diet) in determining the occurrence of an increase in obesity. There are many typical diets, such as the standard and western diets. Brown rice is rich in antioxidants, nutrients, and other chemical elements that benefit a nutritious diet. Many countries’ national dietary guidelines encourage eating brown rice to prevent cardiovascular disease, obesity, and type 2 diabetes [2]. The production of biofunctional materials will increase the contents of the following components: bioactive compounds, such as aminobutyric acid (GABA) [3], dietary fiber, magnesium, free amino acids, potassium, zinc, inositols, tocotrienols, and ferulic acid [4]. However, white rice continues to be generally preferred in rice-eating countries. Across cultures, brown rice was limited by several factors, including its sticky and hard consistency after processing. Since most people consume white rice, some Asians have now started preferring colored/pigmented rice varieties, including black, brown, purple-black rice, and red and reddish brown, due to their various health benefits [2,5]. Whole-grain rice that has been dyed, such as red and black/purple pericarps, is known as pigmented rice. White rice makes up a sizable portion of the calories people consume in Asia and Africa, yet it has less nutritional value when compared to rice variants that are colored. Cereals are an excellent alternative to milk because they stimulate probiotic bacterial growth and resist bile. Brown rice is commonly associated with fermenting species of *Lactobacillus, Streptococcus, Pediococcus*, *Bacillus,* and *Leuconostoc* [6]. Habitually, the two primary reasons for fermenting rice products are (i) to prevent the growth of harmful microbes and (ii) to enhance the flavor, aroma, and texture of the food [7].

Solid-state fermentation trays are the most popular and economical trays. These metal trays with covers hold solid substrate and inoculum and are loosely filled to allow mycelium or other microbes to develop uniformly. Metabolite synthesis depends on microbial growth, and solid-state fermentation (SSF) enhances the process [8]. Due to its cost-effectiveness and eco-friendliness, microbial fermentation is considered the preferable industrial approach for synthesizing bioactive substances over the enzymatic method [9]. Fermented brown rice contains oryzanol and ferulic acid, which have several physiological effects, including lowering serum cholesterol levels [10]. Yi et al. [11] reported that ferulic acid (FA) was the most significant of the four major phenolics in all three brown rice bran samples. Son et al. [12] revealed that dietary FA increased intestinal lipid excretion, lipogenic enzyme activity, and glucose metabolism in HFD mice, suppressing hyperglycemia. The pathways described above are common to humans and *C. elegans* [13]. Activating 112 genes increases body fat in *C. elegans*, whereas inactivating 305 genes lowers fat [14]. In this scenario, *C. elegans* fat deposition, based on lipases, also found in humans, released triglycerides into lipid droplets. It is also possible to observe fat accumulation by employing lipid affinity dyes, which makes it much easier to classify fat deposits in *C. elegans* [15]. Brown rice materials and probiotic fermentation-produced physiologically active compounds may synergistically promote health. Thus, fermented foods across the world provide flavor, aroma, and nutrition to human diets, and the microbes and metabolites of different fermentations have a significant impact on human health.

Our study aimed to meet the needs of rice producers and consumers by assessing phytoconstituents and radicals in brown rice: (1) to examine the antioxidant and antiobesity characteristics of different raw brown rice (BR) over fermented brown (FBR) rice; (2) to determine ferulic acid in raw and fermented samples using HPLC; and (3) to analyze the antiobesity efficacy of the selected FBR (MNL5 FBR) sample in the *C.elegans* model.

## 2. Materials and Methods

### 2.1. Chemicals and Culture Media

De Man, Rogosa, and Sharpe Broth (MRS broth), Tryptic Soy Broth (MB Cell, KisanBio Co., Ltd., Seoul, South Korea), Peptone and Agar (BD Difco, Sparks, MD, USA), ethanol, methanol, isopropanol, DMSO, acetonitrile, and nutritional media were acquired from Daejung Chemicals and Metals Co., Ltd., Siheung-si, Gyeonggi-do, South Korea. Sodium chloride (NaCl), monopotassium phosphate, dipotassium phosphate, sodium chloride, sodium dihydrogen phosphate, sodium carbonate, potassium persulfate, sodium citrate, 2,2′-Azino-bis (3-ethyl benzothiazoline-6-sulfonic acid) (ABTS), 2,2-diphenyl-1-picrylhydrazyl (DPPH), methyl cellosolve, Folin–Ciocalteu reagent, triton-X100, Nile red, 5-fluoro-2′-deoxyuridine, β-carotene, quercetin, ferulic acid, gallic acids, p-coumaric acid, caffeic acid, catechin, and curcumin, lipase from porcine pancreas, dimethyl sulfoxide (DMSO), trolox (≥98% purity by HPLC), and butylated hydroxyanisole (BHA) (≥99%) were acquired from Sigma-Aldrich, Seoul, South Korea. PicoSens™ Triglyceride Assay Kit was acquired from BIOMAX, Korea. All of the reagents were of analytical quality.

### 2.2. Bacterial Strains Growth Conditions and Preparation

All bacterial strains except MNL5 were obtained from Life Science Institute, Well-being LS Co., Ltd., Gangneung, Gangwon, Republic of Korea, under the collaborative project work. MNL5 strain was acquired from Kangwon National University’s Department of Biotechnology, Chuncheon, Republic of Korea. The bacterial strains were maintained in glycerol stock, the subculturing was performed using MRS broth, and the bacterial strains were grown at 37 ± 2 °C for 48 h. Further, the bacterial samples were centrifuged at 15 °C at 8000 rpm for 20 min, and the pellet was dried and stored at 4 ± 2 °C and subcultured once before use.

### 2.3. Preparation of Brown Rice and Fermentation

Brown rice (*Oryza sativa* L.) samples (glutanious brown rice, Hyummi) were obtained from the Life Science Institute, Well-being LS Co., Ltd., Gangneung, Gangwon, Republic of Korea. Firstly, brown rice was soaked in water in a 1:1 ratio at 20–25 ± 2 °C for approximately six hours. After washing, water was removed, and samples were autoclaved at 121 °C, 15 min, followed by cooling at 40–42 °C internal product temperature. Brown rice Samples were inoculated with 1.0 × 10^7^ cfu/g lactobacillus powder and 1% preparation of dry brown rice (weight of brown rice before steaming). The strains (*L. fermentum* (LS-21), *L. Plantarum* (LS-65), *L. Plantarum* (LS-651), *L. acidophilus* (LS-803), and *P. acidilactici* were used for fermentation (MNL-5). The tray fermentation method was performed for 24 h at 37 °C. After 24 h of fermentation, FBR was dried using a drying oven (JSON-050, JSR Co., Ltd., Gongju, Republic of Korea) at 60 °C for six hours. After drying, the FBR samples were grounded using an electric crusher into a fine powder and separated via mesh size 20–30 microns. For further experiments, a 1-mg/mL concentration is used throughout after our previous ethanol extraction procedure [3].

### 2.4. Pancreatic Lipase Inhibition Assay

This experiment was conducted on a 96-well plate following the method described earlier [16]. The percentage of lipase inhibition was used to express the results. Percentage lipase inhibition was ere evaluated from fluorescence measurements with and without substrates. The results were expressed as a percentage inhibition of 1 mg/mL of test extracts or standard.

### 2.5. Determination of Total Phenolic Content (TPC)

The total polyphenols (TPC) were evaluated based on the previous method described by Glorybai et al. [17] with some modifications. The TPC content was presented in gallic acid equivalent per 100 g (mg GA/100 g, DW).

### 2.6. Antioxidant Activity

#### 2.6.1. DPPH and ABTS Radical Scavenging Effect

This experiment was conducted on a 96-well plate using the previously described procedure [18]. The data was presented as mg of Trolox equivalent per 100 g sample (mg Trolox Equiv./100 g, DW).

#### 2.6.2. Ferric Reducing Antioxidant Power (FRAP)

As detailed by Xiang et al. [19], the FRAP assay was analyzed with minor alterations. In brief, 0.1 mL of extracts were mixed with 3.9 mL of a FRAP reagent mixed with acetate buffer (50 mL, 0.3 M, pH 3.6), tripyridyl triazine (5 mL, TPTZ) solution (10 mM of TPTZ in 40 mM of HCl), and FeCl_3_ 6 H_2_O (5 mL, 20 mM) and stored for 10 min at 37 °C. Absorbance was measured at a wavelength of 593 nm. These results were presented as mg of Trolox Equiv./100 g, DW.

### 2.7. Ferulic Acid (FA) Detection in Raw and Fermented Samples Using HPLC

Experiments were carried out using a previously described method [20] with an HPLC method with a C18 analytical column (504971) (Merck KGaA, Darmstadt, Germany) with a particle size of 3 μmol/m2, 25 cm × 4.6 mm at 25 ± 2 °C. At an isocratic flow rate of 1.0 mL/min, the mobile phase was methanol and water adjusted to pH 3.0 with orthophosphoric acid 0.1 N (48: 52 *v*/*v*). The injection volume of the sample was 10 µL. FA was observed at 320 nm. The method took 8 min to run, and all experiments were conducted in triplicate.

### 2.8. Maintenance, Synchronization, and Establishment of a Glucose Diet in Caenorhabditis elegans

In this study, *Caenorhabditis elegans* (*C. elegans)* (two types of strains were applied in the study, wild N2 and clone type Daf-2) and *Escherichia coli* OP50 were acquired from the Caenorhabditis Genetics Center (CGC) in Columbia. The nematode was maintained, developed, and synchronized with bleached gravid hermaphrodites (1:1 ratio of NaOH and sodium hypochlorite). In the bleach solution added, vortexed with eggs, and centrifuged at 1500 rpm for 2 min. After adding an equivalent buffer volume, M9 buffer was used to wash the egg pellet (3 g KH_2_PO_4_, 6 g Na_2_HPO_4_, 5 g NaCl, 1 mL 1 M MgSO_4_, H_2_O to 1 L). The L1-stage worms were transferred to plates of nematode growth medium (NGM) containing *E. coli* OP50 to be fed the worms at a temperature of 20 °C. The research objective was to determine if *C. elegans* has a glucose range of 10–15 mmol/L all over its body. Plates of nematode growth medium (NGM) were seeded with 50 µL glucose (18 mg) and FBR at 37 °C. The L4-stage worms were moved to the NGM Petri dish within four hours [21].

#### 2.8.1. Lifespan Assay for Detecting FBR Effects in the *C. elegans* Model

The *C. elegans* lifespan was assessed following our previous protocol (20). In this experiment, 50 µL of OP50 were spread in LB broth on a 60 mm NGM plate (FUdR) for 24 h at 37 °C. The fifty *C. elegans* in the L4 stage were fed an FBR diet and OP50 plates. All plates were incubated at 20 °C, and dead worms were counted every 24 h. After three days, the worms were transferred to fresh NGM plates containing their specified diet.

#### 2.8.2. Impact of FBR *C. elegans* Body Size

The average life span of OP50 worms was calculated in order to assess the impact of glucose on FBR. The Olympus SZ 61 zoom stereomicroscope and HK3.1 CMOS camera were used to photograph individual worms. Using ToupViewTM 3.7 software, the *C. elegans* size assessed along the nematode’s central axis was measured. At least five nematodes have been photographed in at least three different ways.

#### 2.8.3. Analysis of Lipid Deposition in *C. elegans* Using a Fluorescence Microscope

Nile-red labeling procedures were used in lipid reduction studies. The L4 stage of *C. elegans* was treated with liquid M9 and sodium hypochlorite. After that, L4 *C. elegans* were harvested and rinsed in isopropanol (60%) for 5 min. After 30 min, the worms were rinsed twice with 30% ethanol, and approximately ten worms in PBS saline were transferred to a confocal dish (Cat. No: 102350, SPL Life Sciences, South Korea), where they were observed under an inverted fluorescence microscope with a DP74 camera (Olympus CKX53, Tokyo, Japan) [22].

#### 2.8.4. Fluorescence Quantification

It quantifies fluorescence intensity in Image J software. Nile-red staining revealed a predominance of lipids in the gut, with the anterior section of the intestine appearing significantly brighter than its posterior counterpart. The region for fluorescence density measurements was from the gut to the vulva. Five L4 stage *C. elegans* were selected at random for this study. The photographs were captured at 20x resolution for image J assessments.

#### 2.8.5. Determining the Triglycerides

Triglyceride tests (TG) evaluate RBR and FBR diets, which decrease or increase *C. elegans* lipids. The TG test was performed based on the homogenization of *C. elegans* over 50% of the lifespan towards treatments. TG levels were measured based on Triglyceride Colorimetric Assay Kit with manufacturer protocols (10010303, Cayman Chemicals, Ann Arbor, MI, USA) following the manufacturer’s protocol.

### 2.9. Statistical and Software Analysis

The data were analyzed using Microsoft Excel 365. Fluorescence in the nematode was quantified using the Image J software (Version 2.9.0/1.53t). The results were presented as mean ± standard deviation (SD) from the least-triplicate analyses.

## 3. Results

### 3.1. Effect of Brown Rice Fermentation

A total of 15 strains were evaluated for FBR fermentation. Among them, five strains *L. fermentum* (LS-21), *L. plantarum* (LS-65), *L. plantarum* (LS-651), *L. acidophilus* (LS-803), and *P. acidilactici* (MNL-5), exerted high pancreatic lipase inhibitory activity, TPC content, and antioxidant activities. In our previous studies, the MNL5 strain has shown potential fermentation ability with different substrates by increasing bioactive compounds and antiobesity potential [21,22].

### 3.2. Impact of RBR and FBR Extracts on Lipase Inhibition

The percentage of pancreatic lipase activity measured by enzymes and its product formation inhibitory activities are shown in Figure 1. The fermentation process improved the lipase activities of brown rice compared to the raw extract. However, apart from MNL5, FBR > LS-21 > LS-65, and LS-803 also showed enhanced lipase inhibitory activity compared to raw samples (Figure 1). Lipase inhibition is an effective mechanism for reducing triacylglyceride absorption in hypercholesterolemic patients. Seo et al. [23] reported the synergistic lipase inhibitory activity of kefir LAB and polyphenol-rich GSF by modulating intestinal microbiota in the mouse model. Karamac et al. [24] noticed that caffeic acid and ferulic acid were more effective at inhibiting pancreatic lipase than p-coumaric acid.

### 3.3. Total Phenolic Content of RBR and FBR Extracts

Figure 2 shows the total phenolic content of fermented and raw brown rice. In the MNL5 FBR, TPC increased compared to the raw BR. In comparison to raw BR (298.8 ± 4.52 GAE Equiv./100 g, DW), the MNL5 FBR (650.8 ± 5.81 mg GAE Equiv./100 g, DW) has the most significant total phenolic concentration (Table 1). The second enhanced TPC FBR sample was LS-651, followed by LS-65 and LS-803. The total phenolic content is predicted to contribute to the antioxidant effect. Our results indicate that more research is essential for increasing the bioactivity of phenolics in FBR to achieve antioxidant efficacy. Since phenolic substances are limited, cereals usually have esterified linkages to the grain wall matrix [25]. Fermentation is a promising technique for releasing insoluble or bound phenolic molecules, enhancing grain phenols’ low bioavailability. Our current study compared different fermented bacteria and found that MNL5 had a higher total phenolic content than other bacterial strains. Consequently, increasing the bioavailability and bioaccessibility of cereals, such as brown rice, increases their phytoconstituent content [26].

### 3.4. Antioxidant Assay (DPPH, ABTS, and FRAP) of RBR and FBR Extracts

Several mechanisms, including reducing capacity, free lipid peroxidation inhibition, metal ion chelation, and radical scavenging, have been studied to account for the enhanced antioxidant capabilities found in rice extracts [27]. In recent decades, fermentation has been considered an effective way to boost the antioxidant activity of grains. DPPH, ABTS, and FRAP tests determine the antioxidant activity of raw and fermented brown rice with various LABs. The antioxidant values of DPPH, ABTS, and FRAP of raw and FBR samples are presented in Table 1, respectively.

#### The Effect of DPPH, ABTS, and FRAP Scavenging Activity

Absorption spectroscopy is among the most widely used methods for assessing the antioxidant activity of natural materials. The DPPH was maximum in the MNL5 FBR (124.3 ± 1.8 mg Trolox Equiv./100 g, DW), followed by LS-21, LS-65 FBR (107.30 ± 2.81, 100.10 ± 2.74 mg Trolox Equiv./100 g, DW). The raw BR was examined at the lowest values (55.65 ± 1.44 mg Trolox Equiv./100 g, DW).

Consequently, ABTS is vital for evaluating radical scavenging capacity in grains. Additionally, the FRAP assay is often used to measure the antioxidant ability of isolated compounds and biological samples and to test the antioxidant power of plasma. It detects absorbance variations produced by blue iron (II) from iron oxide (III). In the ABTS and FRAP experiments, the same trend was observed with DPPH. ABTS activity was also highest in the MNL5 FBR (130.52 ± 1.99 mg Trolox Equiv./100 g, DW). In ABTS, the minimum activity was found by using raw BR. In the current study, using ABTS and FRAP assays, the same trend was seen with DPPH. ABTS activity was highest in MNL5 FBR (130.52 ± 1.99 mg Trolox Equiv./100 g, DW), and raw BR had the least activity for ABTS.

Likewise, FRAP was greatest in MNL5 FBR (120.16 ± 1.71 mg Trolox Equiv./100 g, DW), followed by LS-21 and LS-65 FBR. Based on the results of tests performed for antioxidants, MNL5 FBR revealed the maximum activity of all samples. These results were higher than earlier reports by IIowefah et al. [28] in fermented BR. This technique of measuring antioxidant activity and bioavailability of phenolics and flavonoids may indicate differences in extract antioxidant activity. The chemical nature of phenolic compounds directly influences their quenching ability. These free radicals were linked with aging, obesity, and diabetes, as well as the breakdown of critical fatty acids; antioxidants aid in preventing the conditions, as mentioned earlier [29]. Similar findings showed that fermented L. reuterii AKT1 brown rice, rich in antioxidant phytonutrients and functional foods with antioxidant and stress-reducing characteristics, could enhance wellness [3].

### 3.5. Optimization of the Detection of Ferulic Acid

In the RBR and FBR samples, ferulic acid was detected to check the efficacy of fermentation on phenolic content (Figure 2). Ferulic acid is well known for its beneficial effects on various metabolic diseases; therefore, this compound was considered in this study. Ferulic acid content was much more abundant after 24 h of fermentation than in the raw sample (Figure A1). Ferulic acid prevents visceral fat and weight gain caused by a high-fat diet by inhibiting serum amylase and lipase activity and blocking adipocyte production of proinflammatory cytokines (TNF- and MCP-1). FA significantly affects lipid levels in the blood, liver, and skeletal muscle, enhancing antioxidant capacity [30]. FA on influencing metabolic diseases of lipid and glucose pathways and revealing its potential intracellular mechanisms for preventing metabolic syndrome [31]. According to Naowaboot et al. [32], FA promotes glucose and lipid homeostasis in HFD-induced obese mice, most probably through modulating the regulation of lipid metabolism and gluconeogenic genes in hepatic tissue. FA enhances antioxidant activity, lowering fat accumulation, body mass, and hyperlipidemia in obese mice [33].

### 3.6. Effects of RBR and FBR Extracts on the Life Span of C. elegans

A high glucose diet may decrease the *C. elegans’* (N2 and Daf-2) lifespan. The mean and maximum longevity of the MNL5 FBR (N2—40.5 ± 1.0%; Daf-2—18.5 ± 1.1%) groups were significantly longer than that of the raw brown rice (N2—22.1 ± 1.0%; Daf-2—10.1 ± 1.1%) group, as shown in Figure 3. The obesity-induced N2 and Daf-2 *C. elegans* consumed OP50 and glucose (PC) groups, and the whole worms were found dead on the 14th and 20th days, respectively. Our findings supported an earlier study that indicated *P. acidilactici* MNL5 might lengthen the longevity of high sugar-induced worms [21]. Many fermented materials have recently claimed that lipid metabolism regulates *C. elegans* longevity by linking apoptosis, embryonic stem cells, and chromosomal factors. Rice kefir activates DAF-16 and could cause a decrease in ROS. As a result, rice kefiran may cause an increase in life span and an enhancement in motility [34]. Hou et al. [35] reported that zymolytic grain extract increased longevity dose-dependently versus controls. In the present study, MNL5 FBR extracts protected against the decreased lifespan produced by hyperglycemia by suppressing fat deposition.

#### Effects of RBR and FBR Extract on Fat Deposition, Triglyceride (TG) Content, and Gene Expression Levels of *C. elegans*

The nematode fat deposition was determined using Nile-red fluorescence and the TG assay. Nile Red is a benzophenoxazone dye that is an uncharged hydrophobic molecule. It is a fluorescent probe for intracellular lipids and proteins with hydrophobic domains (excitation/emission maxima ∼552/636 nm). As shown in Figure 4a, fatter in the RBR treatment compared to the NC treatment suggested that excessive glucose intake was more significant than in the NC treatment, which indicated that excessive glucose intake increased fat accumulation and body size is increased in the worms as a result of their diet (Figure A2). Compared to the model MNL5 FBR, there are fewer fat deposits in the worm’s body. Figure 4b, c showed that FBR dose-dependently decreased TG, and the inhibitory impact was comparable to the finding of Nile-red staining. The *C. elegans* in vivo model was widely used to study the genetic mechanism of fat metabolism. The intestinal cells of *C. elegans* were responsible for metabolism, absorption of nutrients, and fat reduction and deposition. Usually, TG enhancement is determined as an endpoint while measuring food intake and energy expenditure. Wang et al. [36] revealed that on a cellular level, certain phenolic substances could inhibit adipose proliferation and triglyceride formation while enhancing lipolysis, or the breakdown of triglycerides to glycerol and free fatty acids. Moreover, phenolic substances can affect adipogenesis-related signaling pathways. *C. elegans* has been employed twice to study the antiobesity properties of phenolics.

*C. elegans* is a prominent model for studying the etiology of several disorders and physiological processes, including obesity, longevity, growth, and locomotion [13]. FBR samples were tested using quantitative PCR to evaluate how altered the fat-controlling genes were in *C. elegans*. As depicted in Figure 5, with the consumption of MNL5 FBR, the downregulation of fat-4, fat-5, and fat-6 genes by N2 and Daf-2 model-related lipid metabolism was observed. Raw and PC groups had the most pronounced effects on the above genes, while MNL5 FBR had the most negligible effects.

In *C. elegans*, fat-4, fat-5, and fat-6 are responsible for forming FA desaturase; hence, upregulation of these genes may lead to increased unsaturated fatty acid content. DAF-7 is a transforming growth factor (TGF) receptor created by ASI neurons. It controls the metabolism of lipids and the growth of *C. elegans*. In *C. elegans*, daf-1 and daf-7 alterations and inhibition enhance lipogenesis. Reactive oxygen species typically cause lipid accumulation, which affects health [37]. Moderate-to-strong links among target lipid metabolism genes reveal that MNL5 FBR could suppress lipid accumulation in *C. elegans* and prolong life.

Fermented foods have existed for a long time and are found all over the globe. Due to this, fermented foods that have been consumed traditionally are well-known, common, and diverse. FBR are more palatable, have lower antinutritional impacts, and have enhanced bioactive bioavailability. These results suggest that a FBR dietary supplement or therapeutic agent improves obesity. In addition, we demonstrated that FA could be identified effectively by analyzing FBR metabolites that exhibit beneficial properties in our study. This information is necessary to process the FBR and its products for the food processing industries. Furthermore, new strategies and partnerships between industries, researchers, and relevant organizations are necessary to promote the use of fermented brown rice.

## 4. Conclusions

We found that MNL5-FBR exhibits strong inhibitory activity against pancreatic lipase and is associated with concurrent antioxidant activity in vitro. However, FBR represents a particular source of phenolic compounds. In the *C. elegans* N2 and Daf-2 obese-induced paradigms, our findings indicate that MNL5-FBR has a greater survival rate and is a significant indicator for measuring lipid reduction beyond dietary efforts. MNL5-FBR also increased the amount of ferulic acid, which is directly linked to a longer life span and lower levels of lipids. However, further studies are required to elucidate the control of the fermentation process and to perform an untargeted metabolite analysis. This study will help in a clearer understanding of bacterial synergistic effects. In addition, this study’s findings will be helpful for future research utilizing mice obesity models to examine the impact of FBR on gut microbiota and obesity. Additionally, the current research is part of a larger effort to increase the added value of producing and using fermented brown rice to prevent chronic diseases in humans caused by obesity.

## Figures and Tables

**Figure 1 life-13-00374-f001:**
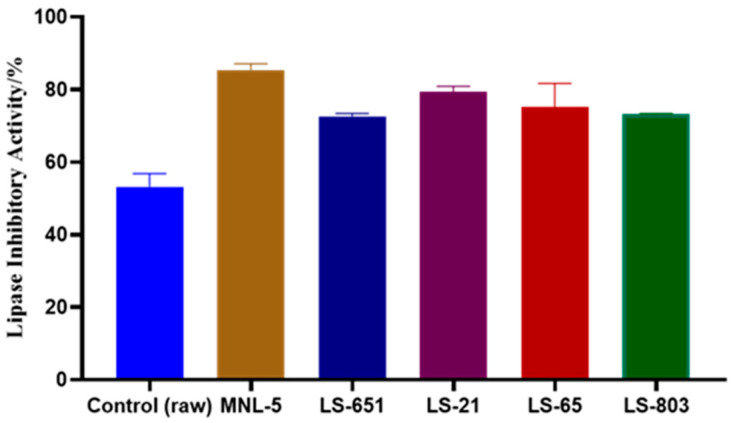
In vitro pancreatic Lipase-inhibiting antiobesity efficacy of several strains of LAB fermented brown rice. All results are shown as the mean ± standard error mean.

**Figure 2 life-13-00374-f002:**
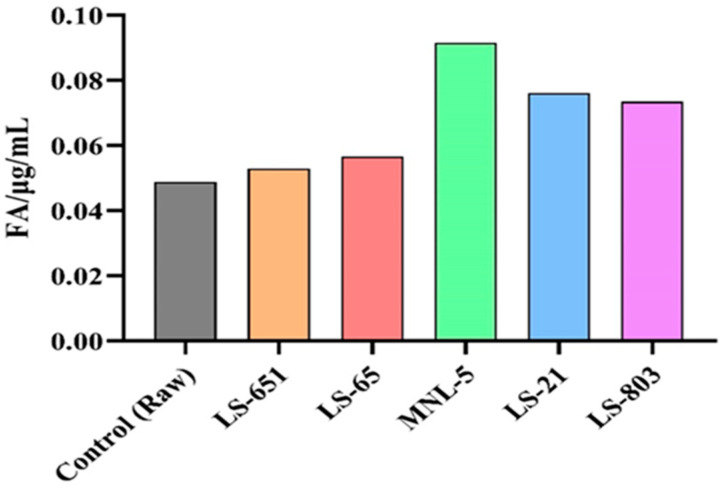
The FA content of RBR and different LABs FBR samples. The data are shown as means SEM, with *p* < 0.05 indicating statistical significance.

**Figure 3 life-13-00374-f003:**
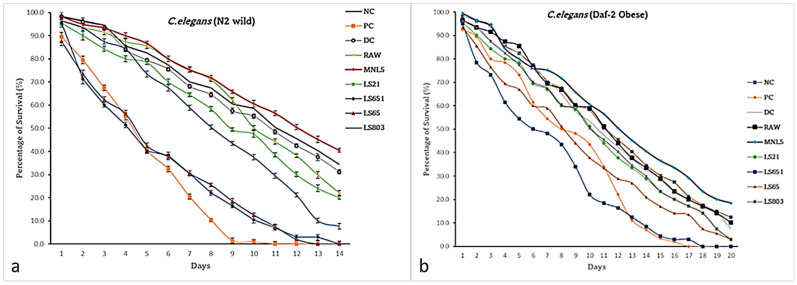
Impact of *C.elegans* lifespan on glucose supplemented RBR and FBR diet. (**a**) N2; (**b**) Daf-2. The different treatments, such as OP50 (NC), OP50 + GLU (*E. coli* OP50 with glucose (positive control), Orlistat + GLU (DC), raw + GLU (raw), FBR + GLU (MNL5, LS21, LS-651, LS-65, LS-803 with glucose) in each test. One hundred worms were fed, repeated, and analyzed by OASIS II. The data are shown as means SEM, with *p* < 0.05 indicating statistical significance.

**Figure 4 life-13-00374-f004:**
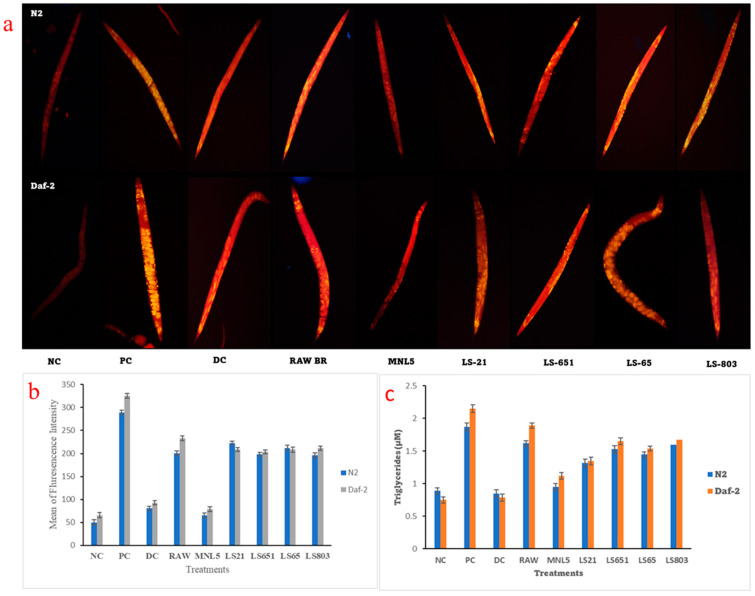
The impact of RBR and FBR diet reduction in *C. elegans* lipid droplets visualized by Nile-red staining method (**a**) N2 and Daf-2 model by fluorescence image (20× magnification); (**b**) mean of fluorescence intensity measured by ImageJ software; (**c**) the lipid and TG levels of worms are shown in a bar chart. The data are shown as means SEM, with *p* < 0.05 indicating statistical significance.

**Figure 5 life-13-00374-f005:**
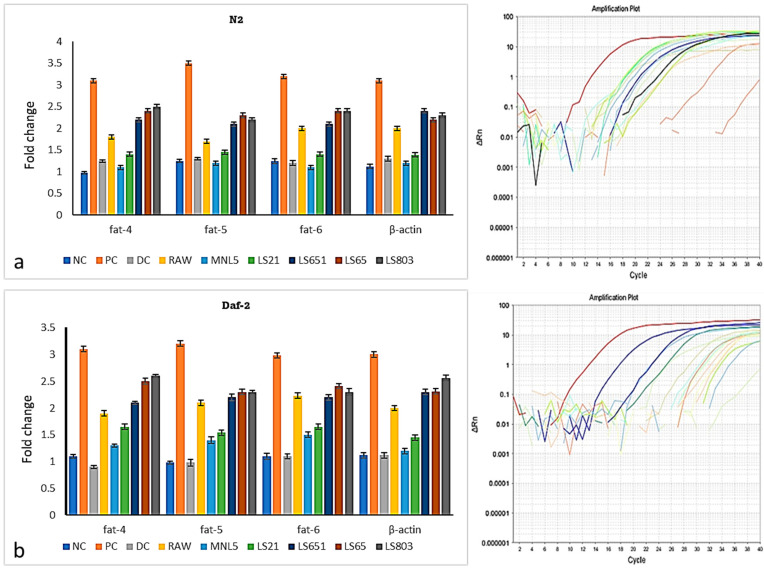
Analysis of gene expression for the RBR and FBR diets on *C. elegans*. (**a**) N2; (**b**) Daf-2; the bar chart shows the gene expression in *C. elegans* treated with RBR and FBR diets. The data are shown as means SEM, with *p* < 0.05 indicating statistical significance.

**Table 1 life-13-00374-t001:** Total phenolic content (TPC) and total antioxidants DPPH, ABTS, and FRAP, of raw and different LABs, fermented brown rice (BR) samples.

S.No	Raw and Fermented Brown Rice Strains	DPPH (mg Trolox Equiv./100 g, DW)	ABTS (mg Trolox Equiv./100 g, DW)	FRAP (mg Trolox Equiv./100 g, DW)	TPC (mg Gallic Acid Equiv./100 g, DW)
1	Control (Raw)	55.65 ± 0.73 ^e^	49.54 ± 0.92 ^e^	58.02 ± 0.96 ^d^	288.83 ± 0.61 ^f^
2	MNL-5	124.40 ± 0.87 ^a^	136.45 ± 1.0 ^a^	120.32 ± 0.64 ^a^	650.79 ± 0.84 ^a^
3	LS-651	68.12 ± 1.01 ^d^	82.45 ± 0.71 ^d^	72.31 ± 0.59 ^c^	569.41 ± 0.73 ^b^
4	LS-21	107.31 ± 0.94 ^b^	108.32 ± 0.83 ^b^	100.30 ± 1.0 ^b^	476.21 ± 1.03 ^e^
5	LS-65	100.10 ± 0.59 ^b^	97.74 ± 0.74 ^b^	95.45 ± 0.88 ^b^	537.45 ± 0.89 ^c^
6	LS-803	90.64 ± 0.99 ^c^	92.31 ± 0.55 ^c^	82.31 ± 0.97 ^c^	493.68 ± 0.51 ^d^

Table 1 control-raw brown rice, MNL5, LS-651, LS-21, LS-65, and LS-803—LAB fermentation BR. The results were represented as the mean and standard deviation of the analyses. Each column’s letters denote (^a^, ^b^, ^c^, ^d^, ^e^, ^f^) statistically significant differences (Tukey and Duncan test *p* ≤ 0.05) DW, dry weight sample.

## Data Availability

Supplementary figure is contained in the Appendix A. Data will be made available on request.

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
