# Peer review of "Antiobesity and Antioxidative Effect of Fermented Brown Rice Using In Vitro with In Vivo Caenorhabditis elegans Model"

_life, 2023, doi:10.3390/life13020374_

Round 1

Reviewer 1 Report

In this manuscript, the authors described the antiobesity and antioxidative effect of fermented brown rice using in vitro with in vivo Caenorhabditis elegans model. I suggest that this is a very interesting paper, and organized well. I just wonder whether authors have considered further applying these fermented brown rice into a special area or not. Thereby, some discussion about economic feasbility should be mentioned. The authors need to provide the all figures with a high resolution (300 × 300 DPI).

Author Response

The authors are grateful for the reviewer's valuable comments, and further, as per the reviewer’s comment, the entire manuscript has been edited to a readable format, as per the reviewer’s valuable suggestion a native English language speaker has edited the entire manuscript.

As per the reviewer’s point, we emphasize the discussion about economic feasibility in the main text of the manuscript.

Fermented brown rice is not only obesity, but our group focusing other health benefits like  Stress reducing (GABA) properties also (https://doi.org/10.3390/antiox10040626; https://doi.org/10.1016/j.foodchem.2022.134747). Some in vitro studies showed anti-diabetes and anti-hypertension also.

In our current study, we collaborate with industrial research (Well-beings LS Co Ltd, Korea) to develop functional food for obesity. In this study, we focused only on obesity.

As per the reviewer’s suggestion, all figures are provided in high resolution.

Reviewer 2 Report

Remarks to the Author:

Kaliyan Barathikannan and colleagues investigated the antiobesity and antioxidative effect of fermented brown rice using in Caenorhabditis elegans. It is an interesting work research, but there are still some things that need to be revised.

Overall, I would support the publication of this study once the authors have addressed a series of changes.

Comments:

1. The full name "Pediococcus acidilactici" should be used for the first time. (line17)

2. “Activating 112 genes increases body fat in C. elegans, whereas inactivating 305 genes lowers fat.”. This sentence does not seem to be known from reference 14. Please add the correct reference.

3. There is an extra space character on line 134 and line 159. The upper and lower case of “p” should be uniform throughout the text, such as “P < 0.05” on line 230 and line 305.

4. It is necessary to add the synchronization step of C. elegans.

5. How many pictures did you used for fluorescence quantification? And what are the emission filter wavelength and excitation wavelength?

6. Because when the C. elegans is placed directly on the slide and the cover slide is covered, it is particularly easy to crush and destroy the structure of the C. elegans. How did you solve the problem? Were the C. elegans transferred onto a 2% agar pad, or did you use any other method?

7. It is necessary to add statistical analysis in Materials and Methods.

Author Response

The authors are grateful for the reviewer's valuable comments, and further, as per the reviewer’s comment, the entire manuscript has been edited to a readable format, as per the reviewer’s valuable suggestion a native English language speaker has edited the entire manuscript.

Comments:

  1. The full name "Pediococcus acidilactici" should be used for the first time. (line17)

Corrected as per the reviewer’s suggestion.

  1. “Activating 112 genes increases body fat in C. elegans, whereas inactivating 305 genes lowers fat.”. This sentence does not seem to be known from reference 14. Please add the correct reference.

References are included in the main text.

  1. There is an extra space character on line 134 and line 159. The upper and lower case of “p” should be uniform throughout the text, such as “P < 0.05” on line 230 and line 305.

Corrected as per the reviewer’s suggestion.

  1. It is necessary to add the synchronization step of C. elegans.

Yes. It is needed for the same age (L4 stage) worms harvested and perform the further experiment and detailed methodology provided in methods.

The nematode was maintained, developed, and synchronized by bleached gravid hermaphrodites (1:1 ratio of NaOH and sodium hypochlorite). In the bleach solution added, vortexed with eggs, and centrifuged at 1500 rpm for 2 min. After adding an equivalent buffer volume, M9 buffer was used to wash the egg pellet (3 g KH2PO4, 6 g Na2HPO4, 5 g NaCl, 1 ml 1 M MgSO4, H2O to 1 L). The L1 stage worms were transferred to plates of nematode growth medium (NGM) containing E. coli OP50 to feed the worms at a temperature of 20°C.

  1. How many pictures did you used for fluorescence quantification? And what are the emission filter wavelength and excitation wavelength?

The triplicate picture was analyzed, and a prominent image was provided in the results. Nile Red is a benzophenoxazone dye that is an uncharged hydrophobic molecule. It is a fluorescent probe for intracellular lipids and proteins with hydrophobic do-mains (excitation/emission maxima ∼552/636 nm).

  1. Because when the C. elegansis placed directly on the slide and the cover slide is covered, it is particularly easy to crush and destroy the structure of the C. elegans. How did you solve the problem? Were the C. elegans transferred onto a 2% agar pad, or did you use any other method?

Yes. We used different methods.

After 30 minutes, the worms were rinsed twice with 30% ethanol, and approximately ten worms with PBS saline were transferred to Confocal Dish (Cat. No: 102350, SPL Life Sciences, South Korea), observed under an inverted fluorescence microscope with a DP74 camera (Olympus CKX53, Tokyo, Japan) [22].

  1. It is necessary to add statistical analysis in Materials and Methods.

Added in that section.
